

**Assessing Bare Ice Albedo Simulated by MAR over the Greenland Ice Sheet (2000-2021) and Implications for Meltwater Production Estimates**

Raf M. Antwerpen[1], Marco Tedesco[1,2], Xavier Fettweis[3], Patrick Alexander[1,2], Willem Jan van de Berg[4]

[1]Lamont-Doherty Earth Observatory, Columbia University, Palisades, NY, USA

[2]NASA Goddard Institute for Space Studies, New York, NY, USA

[3]Department of Geography, University of Liège, Liège, Belgium

[4]Institute for Marine and Atmospheric Research, Utrecht University, Utrecht, The Netherlands

*Correspondence to*: Raf Antwerpen (ra3063@columbia.edu)

**Abstract.** Surface mass loss from the Greenland ice sheet (GrIS) has accelerated over the past decades, mainly due to enhanced surface melting and liquid water runoff in response to atmospheric warming. A large portion of runoff from the GrIS originates from exposure of the darker bare ice in the ablation zone when the overlying snow melts, where surface albedo plays a critical role in modulating the energy available for melting. In this regard, it is imperative to

understand the processes governing albedo variability to accurately project future mass loss from the GrIS. Bare ice albedo is spatially and temporally variable and contingent on non-linear feedbacks and the presence of light-absorbing constituents. An assessment of models aiming at simulating albedo variability and associated impacts on meltwater production is crucial for improving our understanding of the processes governing these feedbacks and, in turn, surface mass

loss from Greenland. Here, we report the results of a comparison of the bare ice extent and albedo simulated by the regional climate model Modèle Atmosphérique Régional (MAR) with satellite imagery from the Moderate Resolution Imaging Spectroradiometer (MODIS) for the GrIS below 70° N. Our findings suggest that MAR overestimates bare ice albedo by 22.8% on average in this area during the 2000-2021 period with respect to the estimates obtained from MODIS. Using an

energy balance model to parameterize meltwater production, we find this bare ice albedo bias can lead to an underestimation of total meltwater production from the bare ice zone below 70° N of 42.8% during the summers of 2000-2021.





## 1 Introduction

Global mean sea level (GMSL) rise has significantly accelerated over the past decades (Chen et al., 2017), in part as a consequence of the acceleration in Greenland ice mass loss (Aschwanden et al., 2019). Ice mass loss from the Greenland ice sheet (GrIS) was one of the largest contributors to GMSL rise in the period 1901-2018 with 17-32% and will likely continue to be so by the end of this century (Fox-Kemper et al., in press). According to Fox-Kemper et al. (in press), the GrIS' contribution to GMSL will constitute ~17% by 2100 for the Shared Socioeconomic Pathway SSP5-8.5 (Riahi et al., 2017). From this point of view, it is imperative to improve model representation of physical processes responsible for ice mass loss to better constrain projections of the future contribution of the GrIS. In this regard, evaluation of climate model outputs vs. observational data can provide insight into the model's ability to represent the physical processes at play and can subsequently highlight regions for model improvement (van den Broeke et al., 2017).

The total mass loss from Greenland can be separated into surface (e.g., runoff) and volume (e.g., calving) losses. For the 2000-2018 period, 55% of Greenland's mass loss originated from surface mass balance (SMB; the balance between accumulation and ablation at the ice sheet surface) and 45% from ice discharge from outlet glaciers (Mouginot et al., 2019). The SMB losses from the GrIS have been increasing since the late 1990s, driven primarily by an increase in melt and subsequent liquid water runoff in response to recent atmospheric warming (van den Broeke et al., 2017). An increase in summer surface air temperatures of ~+2° C has also been observed over Greenland since the early 1990s (Hanna et al., 2012; Box, 2013). This has increased runoff by 40%, while the contribution of changes in precipitation, sublimation, and erosion since the early 1990s are not as substantial (van den Broeke et al., 2016, 2017). The recent increase in observed runoff has also been suggested to be linked to changes in atmospheric circulations around Greenland (Hanna et al., 2014; Tedesco and Fettweis, 2020). (Hanna et al., 2018; McLeod and Mote, 2016) suggest that a significant increase in high summer pressure blocking over Greenland (Greenland Blocking Index, GBI) since the 1990s has been a major driver of the recent increase in surface melt over the GrIS (Fettweis et al., 2013; Tedesco et al., 2016a). Changes in atmospheric circulation promoting enhanced runoff have been characterized by increased shortwave incoming solar radiation (Tedesco and Fettweis, 2020) which, in turn, can increase the absorbed solar




radiation, depending on surface albedo, which can be summarized as the ratio of the energy reflected off a unit area of material over the energy incident over that area.

Besides temperature, albedo also strongly controls surface melting and runoff over the GrIS. More specifically, broadband albedo refers to the wavelength-dependent albedo integrated over the full spectral range, weighted by the contribution of each wavelength. In the case of solar radiation, most of the contribution to broadband albedo comes from the visible wavelengths (300-700nm), since most of the solar energy is concentrated within this band (Liang and Wang, 2020). Albedo is one of the key players in the energy balance for the bare ice of the GrIS, which is exposed when the overlying seasonal snow melts. Snow is characterized by a high albedo of ~0.7-0.85 (Alexander et al., 2014), while bare ice is compacted, densified and aged snow (Wiscombe and Warren, 1980) and is characterized by a low albedo of ~0.55 (Tedstone et al., 2020). Bare ice thus absorbs more solar radiation than snow, increasing the energy available for melting. Even though the bare ice zone encompasses only a small fraction of the GrIS in summer along the margins of the ice sheet, the bare ice zone was responsible for 78% of the runoff from the GrIS in the period 1960-2014 (Steger et al., 2017). Since bare ice albedo strongly controls the amount of runoff from the bare ice zone, it is of key importance in controlling GrIS-wide runoff (Tedesco et al., 2008; van Angelen et al., 2012; Alexander et al., 2019).

Bare ice albedo is spatio-temporally variable at different scales in response to non-linear positive feedbacks between absorbed shortwave radiation and surface melt (Box et al., 2012; Ryan et al., 2019). Therefore, meltwater production does not only depend on timing and persistence of bare ice exposure but also on other modulating factors, such as snowfall, which can cover bare ice with a bright, highly reflective layer of fresh snow. The appearance of bare ice reduces the overall GrIS albedo, leading to more melting from the bare ice zone as well as feeding a positive feedback mechanism which ultimately leads to an acceleration of surface melting (Tedesco et al., 2011).

Bare ice exposure is often associated with the presence of light-absorbing constituents (LACs) on the ice, such as dust, black carbon and organic material (Tedstone et al., 2017). These LACs can reduce surface albedo, (Wientjes et al., 2012; Tedstone et al., 2017), and can subsequently increase melting. Dark bands appear in the bare ice as a consequence of outcropping of ice layers mixed with dust that were deposited in the accumulation zone during the late





Pleistocene and Holocene and, later, transported to the lower ablation zone through ice flow (MacGregor et al., 2020). Black carbon and cryoconite, small cylindrical holes (of a few centimeters to a few meters) in the ice surface containing impurities, have also been found to reduce the albedo of bare ice (Cook et al., 2016; Goelles and Bøggild, 2017). (Wang et al., 2018)

found an abundant presence of supraglacial ice algal blooms in the bare ice zone in the Southwestern GrIS, with a direct link between mineral phosphorus in the ice surface and glacier ice algae biomass (McCutcheon et al., 2021). These light-absorbing constituents reduce the bare ice albedo, further enhancing meltwater production and runoff (Tedesco et al., 2016b; Williamson et al., 2018, 2020; Cook et al., 2020). The difficulty in representing bare ice albedo in climate

models partly originates from a lack of understanding of LACs and may result in a reduced accuracy of runoff projections (Alexander et al., 2014).

In this study, we evaluate the performance of the Modèle Atmosphérique Régional (MAR), a regional climate model especially developed for simulating polar climates (Fettweis et al., 2020), in simulating bare ice extent, bare ice albedo and meltwater production by comparing MAR's

model output with satellite imagery from the Moderate Resolution Imaging Spectroradiometer (MODIS). This study complements the study by (Alexander et al., 2014) who focused on the GrIS-wide albedo. Here, we specifically focus on the bare ice zone below 70° N, which is currently responsible for the majority of meltwater production from the GrIS (Steger et al., 2017). We evaluate MAR on a range of spatial resolutions during June, July and August in 2000-2021. We

use an energy balance model to parameterize the meltwater production and to analyze the effect of a bias in observed and modeled bare ice albedo on estimates of meltwater production.

## 2 Data and methods

### 2.1 The MAR RCM

In this study, we use the Modèle Atmosphérique Régional (MAR) version 3.12 regional

climate model (RCM), which simulates the coupled surface-atmosphere system over the Greenland region (Gallée, 1997; Ridder and Schayes, 1997; Lefebre, 2003; Fettweis et al., 2017) and is forced by reanalysis data or climate model output. Specifically, we force MAR at the lateral boundaries and ocean surface with 6-hourly reanalysis output from ERA5 (Hersbach et al., 2020), produced by the European Centre for Medium-Range Weather Forecasts (ECMWF). The





MARv3.5.2 is discussed and validated over the GrIS (Fettweis et al., 2017), with updates to MARv3.11 discussed in (Fettweis et al., 2021). We point out that in the version we use in this study, the geographical projection has been changed to the Standard Polar Stereographic EPSG 3413 from a previously used custom projection. An issue within the code impacting the snow

temperature at the base of the snowpack has also been corrected in MARv3.12 and it now includes a continuous conversion from rainfall to snowfall from 0° C to -2° C as input for the snow model instead of a fixed value of -1° C (Fettweis, personal comm.). The atmospheric component of the model is described by (Gallée and Schayes, 1994) and the Soil Ice Snow Vegetation Atmosphere Transfer scheme (SISVAT) is used as the surface component of the model (Ridder and Schayes,

1997). The surface model incorporates the snow model CROCUS (Brun et al., 1992), which simulates a set number of layers of snow, ice, or firn with variable thickness and transports energy and mass between each layer. The CROCUS model also provides snow grain properties, which are used to simulate snow albedo. In this study, we run the MAR model over Greenland and produce daily output of variables pertaining to the atmosphere and ice sheet surface in this region

at horizontal spatial resolutions of 6.5, 10, 15, and 20 km.

## 2.2 MODIS data

We obtained MOD09GA Version 6 (Vermote and Wolfe 2015) surface reflectance images over the GrIS from the Moderate Resolution Imaging Spectroradiometer (MODIS) on board NASA's Terra satellite through Google Earth Engine (Gorelick et al. 2017). We use daily summer

(June, July, and August; JJA) images over the period 2000-2021 with a horizontal spatial resolution of 500 m. Corrections have been applied to this product for atmospheric conditions such as aerosols, gasses and Rayleigh scattering (Vermote and Wolfe 2015). We also collected daily snow cover images from MOD10A1 Version 6 from Google Earth Engine over the same period. This product includes a daily cloud mask which we use in this study to flag clouds in the

MOD09GA images. The MOD10A1 product also contains daily albedo values (Hall et al. 2016), though values above a latitude of 70° N may be positively biased due to viewing geometry and large solar zenith angle (Alexander et al., 2014). Consequently, we omit MOD10A1 data above 70ºN from our analysis. We choose the summer 2000-2021 study period to accommodate the observation period of MODIS and to account for the seasonal variability of bare ice exposure on

the GrIS, when surface albedo has the largest impact on SMB (Alexander et al., 2014). To allow



for a daily pixel-by-pixel comparison, we first use GDAL (GDAL/OGR contributors) to reproject the daily MODIS data (MOD09GA and MOD10A1) to the MAR's native projection and simultaneously rescale it to the resolution of each of the MAR products.

## 2.3 Bare ice extent

Bare ice is exposed when the snow blanketing it is removed through surface melting. Most of the areas where bare ice occurs are located along the ablation zone, where ablation is larger than accumulation and the SMB is negative. At the transition between the ablation and accumulation zones lies the equilibrium line altitude (ELA), denoting the elevation where ablation is equal to accumulation and the SMB is 0 (Noël et al., 2019). In order to study the behavior of bare ice

exposure we determine a long-term average ELA from the daily MAR outputs of SMB over the GrIS. We estimate the average ELA at 1679 m a.s.l. for the period 2000-2021 over the entire ice sheet as the 95th percentile value of the elevation values in the ablation zone. Taking the 95th percentile of the long-term average values supports the omission of sporadically high ablation cell detections and provides a conservative estimate of the ELA. Then, we constrain bare ice as

simulated by MAR to cells below the long-term average ELA. In this study, we assume bare ice to be modeled by MAR when the following conditions are met: 1) snow is absent (i.e. snow depth is 0 m) and 2) the average density in the top 1 m exceeds 907 kg/m$^3$. A thin layer of fresh snow (300 kg/m$^3$ in MAR) could cover the ice (920 kg/m$^3$ in MAR) following a brief snowfall event. Solar radiation will not attenuate much in a thin layer of snow and will thus not significantly affect

absorption into the underlying ice (Warren et al., 2006). A thin layer of fresh snow will lower the density of the top layer, however. Therefore, setting a lower limit of 907 kg/m$^3$ for the average density of the top 1 m allows for 2 cm of fresh snow to cover the ice, while also allowing the cell to still be detected as bare ice and not as snow. Taking the average density also ensures that ice lenses are not detected as bare ice. We use the ice mask and digital elevation model of the GrIS as

described by the Greenland Ice Mapping Project (Howat et al., 2014) to select areas where the ice sheet is present (vs. where land is present) and to produce the satellite-derived extent and elevation of the GrIS. We extract the satellite-derived bare ice extent (BIE) on the GrIS by applying an upper threshold of 0.6 to band 2 (841-876 nm) in the MOD09GA product (Shimada et al., 2016). Following the same study, we define pixels with reflectance values above 0.6 in band 2 as snow.



We define the annual maximum BIE as the area covered by those pixels that are detected as bare ice for a minimum of 10% of the observed days in JJA in one year. The aim of this is providing a conservative estimate of bare ice extent while ensuring omission of sporadic and erroneous bare ice detections by MODIS, such as superimposed ice and meltwater lakes and

streams. We define a lower estimate of the MODIS-derived annual maximum BIE as the area covered by the pixels below the long-term average ELA that are detected as bare ice for a minimum of 10% of the observed days in JJA. We also define an upper estimate of the annual maximum BIE, which includes: 1) the area found for lower estimate and 2) the area covered by the pixels that are flagged as clouds in MOD10A1 for a minimum of 90% of the observed days in JJA. For

pixels that are covered by clouds for more than 90% of the observed days in JJA, the view of the surface of the GrIS is obstructed to such an extended degree that bare ice cannot be detected for more than 10% of the observed days in JJA. This automatically excludes them as a possible bare ice pixel, leading to potentially missed bare ice area.

We use forecast verification to quantify MAR's ability to simulate bare ice vs. snow. We

assess MAR's forecast quality by examining the statistical characteristics of the dichotomous categorical forecasts of bare ice (true) vs. snow (false) as compared with observations from MODIS. The frequency of forecasts and observations of both bare ice and of snow are listed in a contingency table. To assess MAR's performance in simulating bare ice, we use Frequency Bias Index, Gilbert Skill Score, True Skill Statistic, and Heidke Skill Score. These statistics indicate a

perfect forecaster with a score of 1 and a random forecaster with a score of 0 (Wilks, 2011).

### 2.4 Bare ice albedo

We evaluate the performance of MAR in simulating bare ice albedo by comparing MAR's modeled albedo values with the albedo values observed by MODIS on the overlapping BIE. The output from MAR contains daily albedo values over the entire GrIS. The bare ice albedo scheme

in MARv2 originally consisted of simply assigning a fixed value of 0.55 to bare ice albedo. Nevertheless, the improved MARv3 we use in this study simulates bare ice albedo as a function of accumulated surface water height and slope of the ice sheet, following an exponential relation between pure bare ice albedo and water albedo (Alexander et al., 2014). MAR includes lower and upper boundaries for the bare ice albedo of 0.5 and 0.55. However, many bare ice albedo values





lower than 0.5 have been observed by MODIS and by some PROMICE automatic weather stations (Tedesco et al., 2016b). Such low albedo values are also the result of LACs on the bare ice, which are not taken into consideration in the MARv3 bare ice albedo scheme. Low albedo values can also arise from accumulated meltwater on the surface of the ice in the form of streams and lakes,

but the relative effect of meltwater has been estimated to be smaller than that of impurities (Ryan et al., 2018).

### 2.5 Meltwater production

We use an energy balance model to parameterize the energy available for meltwater production over the bare ice zone and to isolate the effect of albedo on meltwater production

estimates from the bare ice zone below 70° N. Following (Pellicciotti et al., 2008), we parameterize the energy available for meltwater production as:

$$ME = (a \cdot (1 - \alpha) \cdot SW_{down} + b \cdot LW_{net} + c \cdot SHF + d \cdot LHF) / (\rho_w \cdot L_m), \qquad (1)$$

with daily values for meltwater production (ME), albedo ($\alpha$), downward shortwave radiation ($SW_{down}$), net longwave radiation ($LW_{net}$), and sensible and latent heat fluxes (SHF and LHF)

simulated by MAR. The numerator on the right hand side is equal to the energy available for melt. Dividing by density ($\rho_w = 1000$ kg/m$^3$) and the latent heat of fusion of water ($L_m = 3.34 \cdot 10^5$ J/kg) gives the potential meltwater production in mmWE/day. For this purpose, we use the MAR outputs generated at a horizontal spatial resolution of 6.5 km. We determine the parameters $a$, $b$, $c$, and $d$ by finding the minimum of this unconstrained multivariable function on a daily basis. Figure 1

shows the linear regression between the parameterized meltwater production and the meltwater production simulated by MAR, with an R$^2$ of 0.92. As seen in Figure 1, the parameterization tends to underestimate meltwater production slightly relative to the one simulated by MAR. This could be due to the fact that MAR calculates meltwater production every minute and the parameterization calculates melt only once per day since it uses daily MAR output. Moreover, the feedbacks

between air and surface processes are not captured in the parametrization scheme. Lastly, days with melt occurring only during a part of the day occur at the beginning and end of the melt season with the parameters ($a$, $b$, $c$, and $d$) not being able to fully account for this variation. The fractional contribution to meltwater production of each constituent in Equation 1 are calculated by





multiplying each parameter with the respective net energy flux and dividing by the total meltwater production on a daily basis.

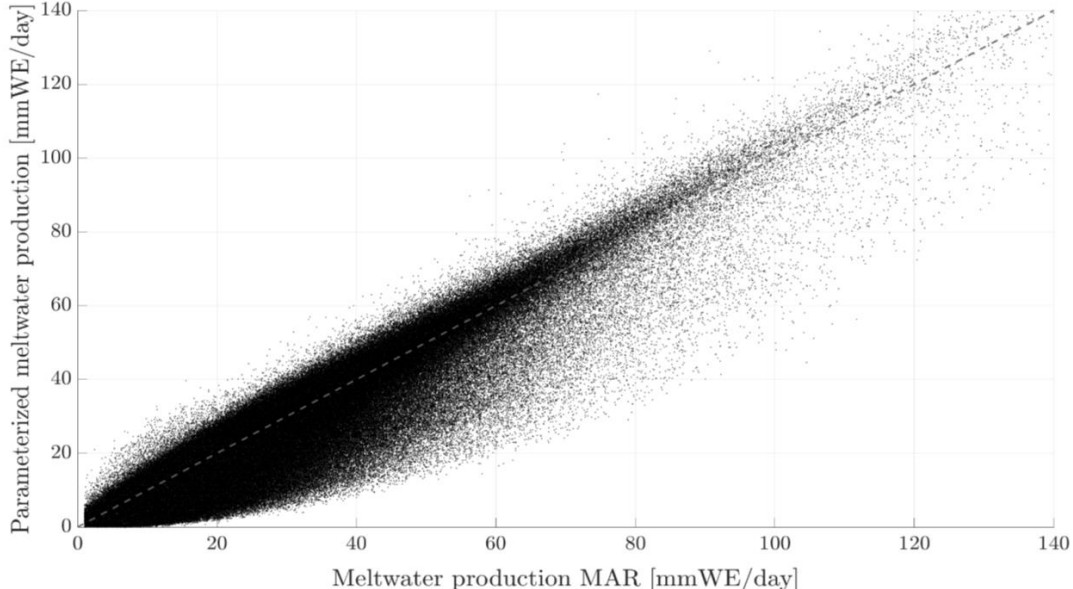

**Figure 1: Linear regression of daily meltwater production simulated by MAR and parameterized meltwater production using modeled albedo.**

We calculate daily meltwater production estimates with the meltwater production parameterization twice, using the same values for the coefficients, once using the albedo modeled by MAR and once using the albedo observed by MODIS. The goal of this is to isolate the effect of bare ice albedo on meltwater production. As a reminder, we exclude cells above a latitude of 70° N to account for the potentially reduced accuracy of albedo values in the MOD10A1 product in this region. In order to increase the fairness of the comparison we include only those areas and days where we simultaneously detect bare ice with both MAR and MODIS. Since we are interested in the effect of bare ice albedo on meltwater production, we only include cells that are melting as prescribed by MAR (>1 mmWE/day). This ensures that any absorbed energy fluxes predominantly go into the enthalpy of fusion of ice, i.e. leading to melt, and not into changing the temperature of the ice.





## 3 Results

### 3.1 Bare ice extent

The average number of days when bare ice is exposed during June, July, and August (JJA) for our study period (2000-2021) obtained from MODIS (Fig. 2a) and MAR (Fig. 2b) as well as their difference (Fig. 2c) are shown in Figure 2. The number of bare ice days observed by MODIS show slightly more inland variation, unlike the modeled number of bare ice days which shows higher values along the western and northern margins of the ice sheet.

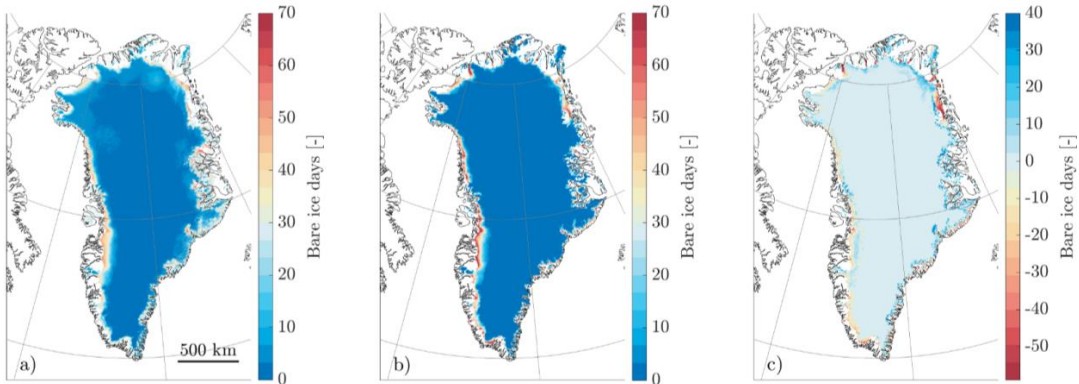

**Figure 2: Number of days bare ice is exposed in JJA, averaged over 2000-2021: a) observed by MODIS on 500 m, b) modeled by MAR on 6.5 km, and c) their difference (MODIS minus MAR) on 6.5 km.**

The inter-annual variability of the maximum extent of the bare ice zone on the GrIS we obtained from the remote sensing and modeled data for the 2000-2021 study period is reported in Figure 3a. The modeled results (lines of different shades of blue, depending on the horizontal spatial resolution) show that MAR agrees well with the general trend of the inter-annual variation in maximum bare ice extent estimated by MODIS. We find $R^2$-values of 0.72, 0.70, 0.65, and 0.65 between the lower MODIS estimate and MAR on 6.5, 10, 15, and 20 km, respectively. In this regard, the MAR output produced at the highest horizontal spatial resolutions (6.5 and 10 km) fit best with the observed inter-annual BIE.





**Figure 3: a) annual maximum bare ice extent in 2000-2021, averaged over JJA and b) seasonal bare ice extent in JJA, averaged over 2000-2021.**





Years with a high maximum BIE, such as 2012 and 2019, correspond to known high-melt years (Tedesco and Fettweis, 2020). This is as expected since warmer temperatures or positive energy balance anomalies lead to more snowmelt, exposing the underlying bare ice. Additionally, exposed bare ice leads to increased absorption of solar radiation, generating higher melt rates (Ryan et al., 2019). In these high-BIE years MAR overestimates the BIE relative to the observations. This indicates that MAR potentially overestimates the amount of snow that is melted away in these years and exposes more bare ice than is actually the case. Some snowfall events may also have been missed by MAR, which would have otherwise reduced the BIE by temporarily covering the ice with a thin layer of fresh snow. Conversely, in years with a low maximum BIE, such as 2006 and 2018, MAR generally underestimates the BIE, suggesting that MAR underestimates the amount of snow that is melted away in these years. This indicates that MAR could be too sensitive to temperature; in colder periods there is not enough snowmelt, in warmer periods there is too much snowmelt. The BIE shows a positive trend during the study period for both MODIS (1,486 km$^2$/yr) and MAR (2,303, 2,082, 1,951, and 2,409 km$^2$/yr on 6.5, 10, 15, and 20 km, respectively). The significantly lower value for the average observed BIE stems from the inclusion of 2021 data, where estimates between observation and model differ vastly. This difference is potentially caused in part by an anomalously high number of clouds over the bare ice zone in mid-August of 2021, obstructing view of the potential BIE which MAR does model as bare ice.

The observed and modeled seasonal BIE exhibit a peak from mid-July through mid-August (Figure 3b). The resolution we use in MAR has minimal effect on the timing and magnitude of the BIE. The cloud-uncertainty for the seasonal observed BIE shows the daily average clouded area over the ablation zone. Hence, it covers a larger relative area than for the annual maximum BIE, where many daily cloud observations cancel each other out. See Methods for more details on this.

The initial modeled BIE is slightly lower than the observed BIE, meaning that the onset of bare ice is delayed in MAR relative to the observed onset. The lower initial modeled BIE may also suggest that MAR misses uncovering of ice by snowdrift which would blow snow into crevasses and holes. In mid-June, the modeled BIE quickly surpasses the minimum observed estimate and in mid-July it surpasses the maximum observed estimate, which includes the clouded area. Again, this could indicate that snowmelt in MAR is too sensitive to temperature. It could also suggest that



MAR is too "eager" to transform firn into bare ice in the lower percolation zone. The dip in late July is in part due to snowfall events over the BIE as simulated by MAR, though the snowfall is not persistent enough to match the BIE up with the observations until mid-August. This suggests that MAR accumulates snow more quickly than what is observed, possibly because snowfall is

overestimated at the end of the season, or because late season snowmelt events are underestimated.

Table 1 shows the contingency table of the dichotomous categorical forecasts and observations of bare ice vs. snow. MAR has a Frequency Bias Index of 0.666, showing that, when bare ice is observed by MODIS, MAR simulates bare ice on the same pixel 66.6% of the time. In other words, MAR underforecasts bare ice exposure. Furthermore, we find values of 0.21, 0.31,

and 0.35 for the Gilbert Skill Score, True Skill Statistic, and Heidke Skill Score, respectively. All three of these statistics show that MAR performs better than a random forecaster (score = 0), but not as well as a perfect forecaster (score = 1) (Wilks, 2011).

**Table 1:** Contingency table of forecasts (MAR) and observations (MODIS) in terms of determining either bare ice or snow. Bare ice is both forecast and observed is a hit (a), bare ice is

forecast but snow is observed is a false alarm (b), snow is forecast but bare ice is observed is a miss (c), snow is both forecast and observed is a correct rejection (d). The sample size is n = a+b+c+d.

|  | **Bare ice observations** | **Snow observations** | **Marginal totals** |
|---|---|---|---|
| **Bare ice forecasts** | a = 1623325 | b = 1036407 | a+b = 2659732 |
| **Snow forecasts** | c = 2367903 | d = 9908829 | c+d = 12276732 |
| **Marginal totals** | a+c = 3991228 | b+d = 10945236 | n = 14936464 |

## 3.2 Bare ice albedo

The average observed bare ice albedo over 2000-2021 exhibits high spatial variability (Figure 4a), with large sections of low albedo values (< 0.4) in the southern region of the GrIS, especially over the dark ice zone in the southwest. The low albedo values in the dark ice zone suggest the presence of abundant LACs, such as black carbon, mineral dust, volcanic ash, cryoconite and ice algal blooms (Tedstone et al., 2017). We observe little to no variability in





average bare ice albedo for MAR (Figure 4b). This is expected from the bare ice albedo scheme in MAR since it does not account for any form of LACs. Albedo values higher than expected for bare ice are detected by MODIS in the northern section of the ice sheet. This is an artifact of the positively biased MOD10A1 product above latitudes of 70° N (Alexander et al., 2014).

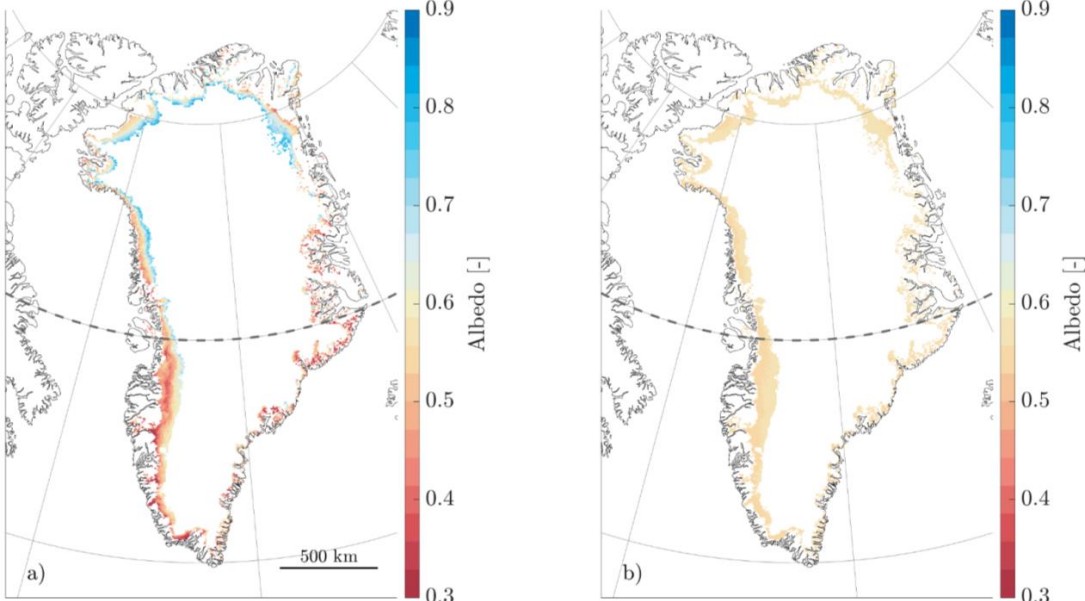

**Figure 4: Maps of bare ice albedo over maximum bare ice extent averaged over JJA in 2000-2021: a) observed by MODIS and b) modeled by MAR. The dashed line denotes 70° N.**

The annual average observed albedo over the overlapping bare ice extent below 70° N shows significant variability throughout the study period for all resolutions, with little to no
10   variability between MAR resolutions (Figure 5a). The variability in annual average modeled bare ice albedo is negligible and remains almost constant at around 0.55, on average 0.12 (or 27.5%) higher than the average of 0.43 observed by MODIS. The observed bare ice albedo shows an average trend of -0.015 ± 0.0025 per decade. We have given each resolution equal weight in the calculations of the means and trends.





**Figure 5: a) annual bare ice albedo in 2000-2021, averaged over JJA and b) seasonal bare ice albedo in JJA, averaged over 2000-2021.**

5       The seasonal average observed bare ice albedo below 70° N exhibits significant changes throughout the JJA season (Figure 5b). The observed albedo (on 15 and 20 km resolution) in June




shows high variability, which is a result of the minimal bare ice extent on the GrIS in this period and the even smaller bare ice extent overlapping between model and observation. Hence, for the first ~20 days of June only a small number of cells is available (less than 10 cells per day) from which to determine the average bare ice albedo, making anomalous values weigh more heavily in

the average. The observed albedo declines rapidly in June and reaches a sustained minimum of ~0.41 from early July until early August. Throughout the season, MAR overestimates bare ice albedo relative to the observations fairly constantly at ~0.55. The difference between observed and modeled albedo originates in part from the missing representation of LACs in the bare ice albedo scheme in MAR (Fettweis et al., 2017). The dipping trend in observed albedo suggests an increase

in spatial distribution or intensification of LAC concentrations. Algal blooms flourish and multiply (Wang et al., 2018, 2020). Holocene dust and black carbon are exposed through melting deeper and older ice layers. Holocene dust and black carbon are continuously being outcropped through melting of deeper and older ice layers and can accumulate on the surface of the ice (Doherty et al., 2013). Significant aeolian depositions of black carbon have also been observed (Goelles and

Bøggild, 2017). Volcanic ash will only play only a minor role in lowering the albedo as it is distributed only in short time intervals during volcanic eruptions. Despite its dark surface, cryoconite has been shown to play a minor role in lowering albedo due to its sparse spatial distribution (Ryan et al., 2018). A part of the seasonal decrease in bare ice albedo also arises from accumulated surface meltwater on the bare ice, which may be misrepresented in MAR.

**3.3 Meltwater production**

Figure 6a and 6b show annual and seasonal averages, respectively, of the fractional contributions of each of the constituents in the meltwater production parameterization, where we use albedo values from MODIS in the shortwave radiation term. The shortwave radiation term is consistently the largest term contributing to meltwater production owing to the low albedo of bare

ice and the long days during boreal summer. Longwave radiation contributes to a net loss of heat in general during the study period, releasing more energy from the surface of the ice than it absorbs. This leads to a net negative contribution to meltwater production. Sensible heat flux contributes a small but rather constant fraction to meltwater production. The amount of melt induced by latent heat flux is very small and often negative, meaning that the latent heat flux on average results more

in evaporation and sublimation than condensation and deposition.





**Figure 6: Fractional contributions of energy fluxes to meltwater production over the bare ice zone: a) annual averages for 2000-2021 and b) seasonal averages for JJA.**

5          We quantify the effect of the bare ice albedo bias by determining the meltwater production in two scenarios: once with the observed albedo and once with the modeled albedo as input to the




meltwater production parameterization. We use the ratio of the daily averages of these two meltwater production estimates to isolate the effect of bare ice albedo on meltwater production from the bare ice zone below 70° N on a daily basis (Figure 7). A positive ratio indicates that using the modeled albedo in the parameterization results in an underestimation of the average meltwater

production on that day, compared to using the observed albedo.

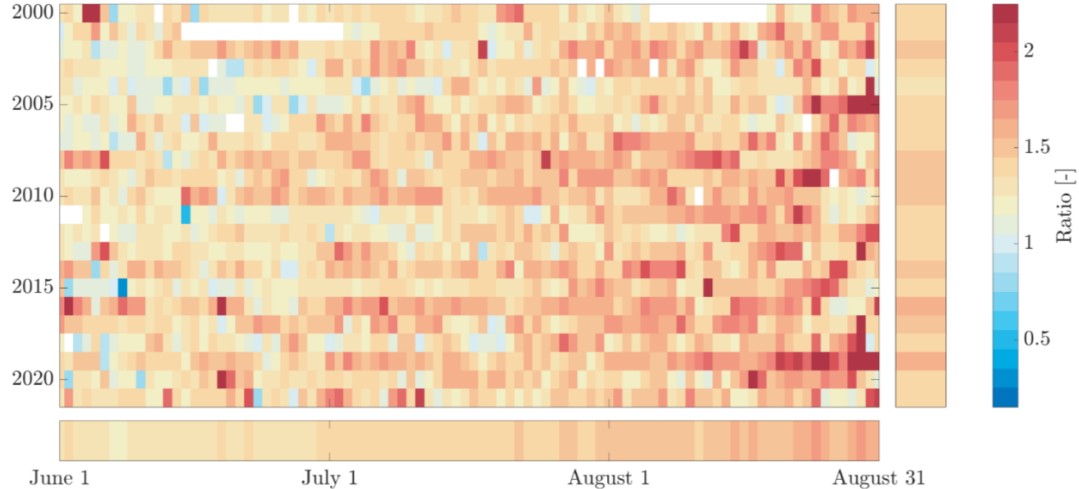

**Figure 7: Ratio of daily parameterized meltwater production with observed albedo to parameterized meltwater production with modeled albedo. The vertical and horizontal bars show annual and seasonal averages.**

We observe a strong increasing seasonal trend in the average seasonal ratio of 0.0245 per week from June through August with seasonal average ratios for June, July and August of 1.32, 1.42, and 1.54, respectively, indicating a seasonally increasing underestimation of parameterized meltwater production using the modeled albedo. The increase in the seasonal average ratio of meltwater production indicates that the spatial distribution and intensity of LAC concentrations on

bare ice could be increasing during the season, reaching peak values in late August. Increasing amounts of accumulated surface meltwater could also be a cause of the increasing trend in the ratio. As shown earlier, the daily bare ice extent significantly decreases after mid-August. This means there is a smaller area over which meltwater production is calculated, increasing its variability. The seasonal average reaches a low of 1.19 on June 7, and a peak of 1.75 on August

24. The meltwater production ratio exhibits significant daily variability throughout the study





period. Daily ratios in June vary from 0.34 to 3.04, though these extreme ratios are sporadic. Since the overlapping bare ice extent between observation and model is small in early June, this period shows extreme variability in observed bare ice albedo and, thus, in meltwater production estimates. Hence, strong conclusions cannot be drawn for early June. The minimum and maximum daily

ratios in July are 0.81 and 2.11, respectively. August exhibits numerous extremely high daily ratios, especially in late August, with minimum and maximum ratios of 0.98 and 3.09, respectively. The average annual ratio exhibits an increasing trend of 0.043 per decade from 2000 through 2021, indicating that parameterized meltwater production from the bare ice zone below 70° N is being increasingly underestimated when using modeled albedo versus observed albedo. One explanation

for this could be an annually increasing amount of LACs that are deposited onto and exposed in the bare ice zone during the summer. Increasing temperatures and accumulated surface meltwater could also create more favorable conditions for algal bloom growth. The minimum and maximum annual average ratios are 1.26 and 1.58, in 2004 and 2019, respectively. Averaged over the bare ice zone below 70° N and the entire study period, the MAR-derived meltwater production using

the modeled albedo could be underestimated by 42.8%, owing to an average overestimation of modeled bare ice albedo of 22.8%. The meltwater productions using observed and modeled albedo have a correlation coefficient of $R^2 = 0.60$.

In addition to the examination of time series, we average the meltwater production ratios over the study period and map them onto the bare ice extent (Figure 8). We find high ratios over

the dark ice zone, which is as expected from the high LAC concentrations in this area (Wang et al., 2020). Moreover, we find values close to or slightly lower than 1 higher up in the ablation zone in the southwest. In this region, meltwater production estimates are close to 0 using both observed and modeled albedo. Melting also occurs significantly less frequently with increasing elevation in this region. Hence, a small difference in albedo can result in a large percentage change of simulated

meltwater production. We find extremely high ratios along the eastern margin where the observed albedo is significantly lower than the ~0.55 simulated by MAR (Figure 4). The large albedo differences and low number of melting days in this area makes meltwater production estimates more variable.



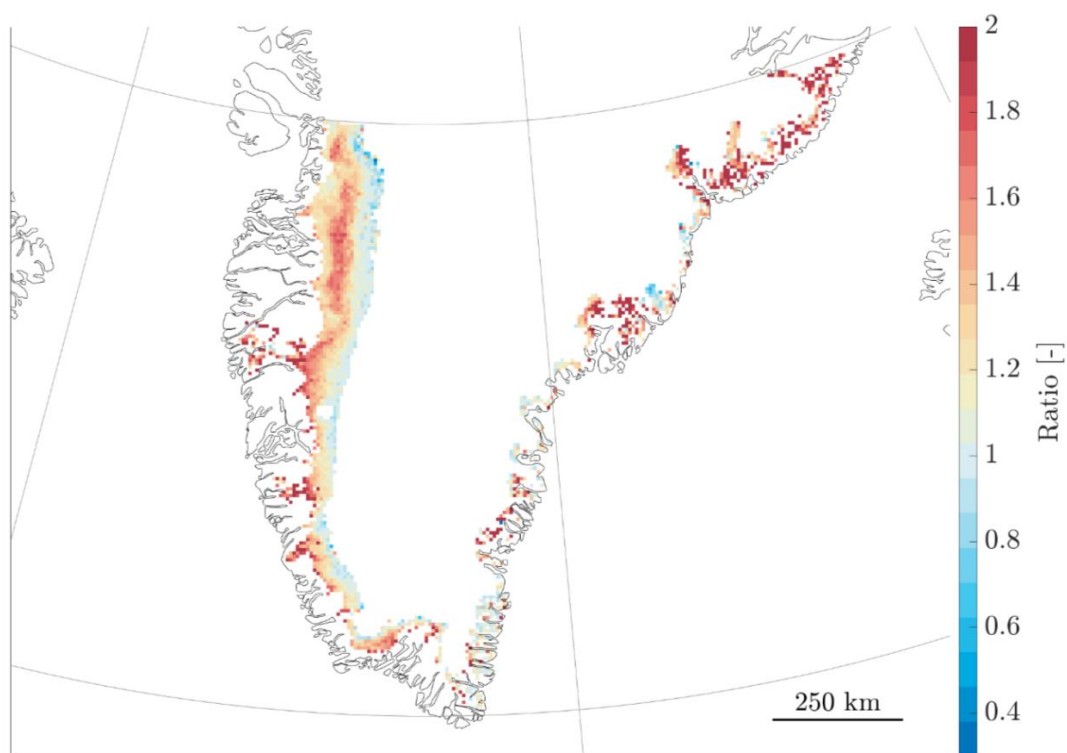

**Figure 8: Ratio of parameterized meltwater production with observed albedo to parameterized meltwater production with modeled albedo, averaged over the entire study period.**

**4. Discussion**

An analysis by (Ryan et al., 2018) on sources of spatial albedo variability along a transect in the dark ice zone, perpendicular to the ice margin, found that 73% of spatial albedo variability can be attributed to LACs on the surface of bare ice; a mixture of algae, dust and black carbon. Only 15% of the spatial variability of albedo is explained by accumulated surface meltwater in
rivers, streams, ponds and lakes. Crevasses are responsible for 12% of the observed albedo variability. Despite the very low albedo of cryoconite, due to its low abundance it accounts for only 0.6% of the albedo variability. Moreover, accumulated surface meltwater may act as a distributor of LACs; a small change in accumulated surface meltwater may thus result in larger albedo changes than merely the added surface water. Granted, the analysis by (Ryan et al., 2018)
only holds for one transect covering 12.5 km$^2$ on August 6, 2014, and may not necessarily be



representative of the entire bare ice area . A modeling study from (Goelles and Bøggild, 2017) suggests that melt-out of englacial black carbon and dust are responsible for most of the LAC-driven meltwater production. Atmospheric deposition of black carbon and dust has a significantly lower effect on meltwater production in their study. These results hold for the location of an AWS
(KAN_M) at 1270 m a.s.l, in the dark ice zone, for 2010-2015. It should be noted that biological activity is not included in their model, so they could be interpreting some albedo changes due to biological activity as effects from black carbon and dust. It thus remains unclear how the effect of algae relates to the effect of dust and black carbon on bare ice albedo and meltwater production. Though, qualitatively, we assume the conclusions drawn by (Goelles and Bøggild, 2017; Ryan et
al., 2018) hold for the albedo differences between model and observations we find in our analysis.

We want to emphasize that our results only hold for the bare ice extent below 70° N that is simultaneously observed by MODIS and modeled by MAR. Since the MOD10A1 product may be less reliable above 70° N, we exclude this region from our analysis. This means that the results shown in this study cannot be extrapolated to the northern half of the GrIS, though we expect the
physical processes to be fairly similar over the entire GrIS. An improved understanding of errors in satellite-derived albedo measurements, including additional high-quality in situ measurements, would be useful for properly analyzing the effects of albedo on meltwater production above 70° N.

The ratios we mention in Section 3.3 pertain to the parameterized meltwater production
using observed albedo and modeled albedo. No direct conclusions can thus be drawn on the performance of MAR in simulating meltwater production over the bare ice zone. However, in the Methods section we show that the parameterized meltwater production using the modeled albedo and the original meltwater production in MAR have a very high correlation ($R^2 = 0.92$). We therefore believe that our conclusions are likely transferable and applicable to the performance of
MAR in simulating meltwater production.

It is also important to note that the SMB simulated by MAR compares very well with SMB observations from the Programme for Monitoring the Greenland Ice Sheet (PROMICE) on average over the entire GrIS (Fettweis et al., 2020). This suggests that the effects on meltwater production of a too low bare ice albedo through absorption of shortwave radiation might be compensated by





other energy fluxes (LWnet, SHF, LHF) in the energy balance equation of MAR over the bare ice area. This is discussed in (Fettweis et al., 2017), who highlighted an overestimation of albedo and downward shortwave radiation but an underestimation of downward longwave radiation. However, MAR underestimates melt at AWS locations in the ablation zone where melt is larger

than 2 mWE/yr (Fettweis et al., 2020), suggesting that at these locations a lower bare ice albedo would improve comparison of modeled SMB with observations from PROMICE. At these locations, the SMB modeled by the Regional Atmospheric Climate Model (RACMO2.3p2) compares better with PROMICE than the SMB from MAR does, most likely since RACMO integrates MODIS albedo into their model (Noël et al., 2019).

**5. Conclusions**

Using remote sensing observations and an energy balance model to parameterize meltwater production, we analyze the performance of the regional climate model MAR in simulating the spatio-temporal variability of the bare ice extent and albedo as observed by MODIS. We have shown that MAR performs reasonably well in simulating the bare ice extent on an annual basis.

Despite the similarities in maximum annual bare ice extent, MAR overestimates the daily bare ice extent during peak bare ice season from mid-July through mid-August. We also conclude that MAR overestimates bare ice albedo below 70° N on average by 22.8% during the study period. This complements and builds further on a study by Alexander et al. 2014, who analyzed surface albedo over the entire GrIS. We advocate that this significant difference in bare ice albedo arises

in substantial part from the lack of LAC representation in MAR's bare ice albedo scheme. A misrepresentation of accumulated surface meltwater on bare ice in MAR may also in part cause the difference between observed and modeled bare ice albedo. Using the meltwater production parameterization, we isolate the effect of the bias in observed and modeled bare ice albedo on the meltwater production from the bare ice zone below 70° N. We find that, using the modeled albedo

in the parameterization, meltwater production is underestimated on average by 42.8% during the study period. The underestimation of meltwater production increases on average with 2.45% per week from June through August and with 4.3% per decade from 2000 through 2021. The largest discrepancies in meltwater production are located over the dark ice zone, where the highest LAC concentrations are found, and along the eastern margins of the ice sheet, where simulating bare ice

extent is more difficult owing to the steep topography of the fjords and cliffs. Since meltwater





production estimates from MAR and estimates from the parameterization with the modeled albedo are closely linked ($R^2$ = 0.92), we believe that the results pertaining to meltwater production are likely transferable and applicable to MAR's performance in simulating meltwater production.

The results of this study show that research efforts should be directed towards uncovering

the spatial and temporal variability of the distribution and trends of LAC concentrations on bare ice. Regional climate models, such as MAR, should work towards adopting a bare ice albedo scheme that allows for inputting spatially and temporally variable LAC concentrations on bare ice. Radiative transfer models such as the SNow ICe and Aerosol Radiative model SNICAR are being improved to allow for inputting black carbon, brown carbon, dust, ash, and algae with a range of

properties in a variable concentration (Whicker et al. 2021, in press). However, no GrIS-wide, or dark ice zone-wide, quantification of distributions and trends are available yet. Hence, this aspect of LACs on bare ice still has to be parameterized.

As global, and more so Arctic, atmospheric temperature continues to rise, more bare ice will be exposed by melting the snow that usually blankets the bare ice, increasing meltwater

production from the ablation zone of the GrIS. Correctly modeling and predicting bare ice albedo, and in particular LAC concentrations on bare ice, is thus becoming increasingly imperative for proper projections of meltwater production from the GrIS by regional climate models and general circulation models.



**Code and data availability**

The code underlying the processing and analysis described in this paper are available from Raf Antwerpen upon request. The MARv3.12 code and outputs are available at https://www.mar.cnrs.fr. MODIS MOD09GA (https://doi.org/10.5067/MODIS/MOD09GA.006, Vermote and Wolfe, 2015) and MOD10A1 (https://doi.org/10.5067/MODIS/MOD10A1.006, Hall et al. 2016) data are available from Google Earth Engine (https://earthengine.google.com, Gorelick et al. 2017).

**Author contributions**

RA, MT, PA and WB designed the study. XF ran MARv3.12 simulations and provided the outputs. RA processed the data, performed the analysis, generated the figures and wrote the manuscript. All authors discussed the results and contributed to the final manuscript.

**Competing interests**

The authors declare they have no conflict of interest.

**Acknowledgements**

Raf Antwerpen would like to acknowledge financial support by National Science Foundation ANS #1713072, National Science Foundation PLR-1603331, NASA MAP #80NSSC17K0351, NASA #NNX17AH04G, and the Heising-Simons Foundation.

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
