# Peer review of "Assessing Bare Ice Albedo Simulated by MAR over the Greenland Ice Sheet (2000-2021) and Implications for Meltwater Production Estimates"

_EGUsphere, 2022_

## Referee Comment (RC1)

Review of

**Assessing Bare Ice Albedo Simulated by MAR over the Greenland Ice Sheet (2000-2021) and Implications for Meltwater Production Estimates**
by Antwerpen et al.

https://doi.org/10.5194/egusphere-2022-37

**General comments:**

The paper presents results from a comparison of the bare ice extent and albedo simulated by the regional climate model MAR and derived from MODIS satellite imagery. Additionally, an energy balance model is used to estimate melt water production. Authors limit their investigations to the GrIS below 70°N in the time period 2000-2021.

Overall, the differences between model results and observations are quite substantial. The albedo parameterization scheme in MAR is identified as a major source of these differences. It is underlined that the impact of light-absorbing constituents in ice and accumulated surface melt water on the GrIS is not properly accounted for in the albedo parameterization scheme.
As discussed in the paper, already earlier evaluations of MAR results pointed to these deficiencies. Hence, what can we really learn from the paper presented here? What is the benefit for the modelling community?

I miss sensitivity studies with respect to the albedo parameterization scheme and it would have been very helpful to implement and test at least simple schemes that account for LAC and melt water signatures on the ice sheet. To know the impact of different parameterization schemes on bare ice extent and melt water production and a comparison to observations would be really interesting and very helpful.

The subject is appropriate for EGUsphere.
The title reflects the content of the paper, the abstract provides a complete summary and the paper is generally well structured.
The review of existing published work is good, the number of references is appropriate.
Overall, figures and tables are clear and their captions self-explanatory.
The use of the English language is very good.

---

## Referee Comment (RC2)

**Comments on "Assessing Bare Ice Albedo Simulated by MAR over the Greenland Ice Sheet (2000–2021) and Implications for Meltwater Production Estimates" by Raf Antwerpen et al.**

**1  Summary**

The authors assess the performances of the Modèle Atmosphérique Régional (MAR) in simulating Greenland bare ice albedo by comparing MAR results over the period 2000-2021 with albedo and bare ice extent retrievals from MODIS optical imagery. Spatial and temporal similarities and deviations in bare ice extent and bare ice albedo between MAR and MODIS retrievals are first presented and discussed. An energy balance model is then used to parameterize meltwater production and present the consequences of significant deviations between modeled and observed bare ice albedo in runoff estimation.

The authors have presented a well-written and coherent manuscript on a topic and scope that suits well for The Cryosphere. Reporting on and understanding the performances of regional climate models in simulating bare ice processes and their consequences is indeed a key step to move towards an improved representation of this complex but highly important area of the Greenland ice sheet. By quantifying the deviations between MAR simulations and spaceborne observations, the authors make an essential contribution to the basis of an improved projection of the Greenland ice sheet's future evolution.

Along with several minor comments, I nevertheless here point out a few substantial concerns that are in my opinion essential to address before publication. Several choices made by the authors are I think not motivated and discussed enough to give the reader a clear picture about their limitations and potential implications. While I mostly support those choices, it is currently hard to get a sense about the magnitude of their influence on the conclusions. My main concerns are about the choice of method for bare ice mapping with MODIS data, the use of a spatially- and temporally-averaged ELA and a 1999-2002 static ice mask and DEM. These different points are further discussed in the "General comments" section below.

**2  General comments**

The threshold of 0.6 applied to MODIS band 2 has been determined by Shimada et al, 2016 using one single satellite image analyzed over West Greenland. This threshold is here applied by the authors to data covering the entire ice sheet from 2000 to 2021 (with a focus below 70°N for runoff estimations). It is therefore important to discuss a potential lack of representativity of this threshold at the ice sheet scale, and present potential associated limitations. In this way, important to better understand the sensitivity of the results to this threshold: does a small variation in this value have a significant impact on the results presented here? And if so, to which extent?
In this direction, and as an example and illustration, I used MOD10A1 (1km resolution) data I had locally on hand for the period 2000-2019 and applied the bare-ice-onset albedo value of 0.565 determined in Wehrlé et al. (2021) to quantify the daily bare ice extent. In Wehrlé et al. (2021), the seasonal evolution of bare ice albedo across Greenland has been studied using data from 20 automatic weather stations (105 station years). A bare-ice albedo at ice-ablation onset of 0.565 -used here- has been determined (and called bare-ice-onset albedo). The MOD10A1 data used here is gapless, a given pixel value is fixed to the last cloud-free retrieval until the next cloud-free update is available. The static GIMP mask has been used. The data and method used here are different to those presented in this manuscript but both exercises quantify the same variable and therefore, should ideally show low deviations. As visible in the figure below, large variations are shown compared to MOD09GA with the Shimada et al 2016 threshold, and the bare ice area computed here is in much better agreement with MAR. Making the assumption MAR is underestimating runoff over the whole ice sheet (although only shown below 70°N in this study), one could

suggest this pattern is mainly due to MAR's incapacity to model low-enough bare ice albedo values (Figure 5b of this study) and not because of low performances in bare ice classification. However, a spatial analysis would be needed to make sure that, more than having similar magnitude, both bare ice extents actually correspond to similar areas.

While MAR uncertainty and potential sources of errors are widely discussed which adds a lot of value to the conclusions drawn in this manuscript, there is currently no similar considerations on the observation side of the study. The example presented here supports the real need for a substantial discussion of the observation retrievals and their sensitivity to the methods applied.

[Figure]

Figure 1: Bare ice area determined from MOD10A1 (2000-2019) and a 0.565 threshold on albedo (Wehrlé et al., 2021) on top of Figure 3 presented in this manuscript. Data has been coarsely digitized. The MAR curve has been digitized as an approximation of MAR 6.5km-20km.

As a first step in the mapping of bare ice, the bare ice extent is bounded to the area below the spatially and temporally-averaged ELA to exclude "sporadically high ablation cell detections". How important are these cells? Do they correspond to <1% of the total bare ice extent or more? Addressing those questions would be necessary to get a better sense about the sensitivity of the results to the use of the long-term average ELA (more details in detailed comment P7 L13-14).

The GIMP ice mask and DEM used in different sections of the study to assess results over the 2000-2021 time period have been created from a combination of Landsat-7 and RADARSAT-1 imagery acquired between 1999 and 2002. The ice sheet evolved significantly in 20 years. It is important to discuss if the use of a static ice mask and DEM has an important impact on the different results it is related to, in order to, here again, be able to get the full picture about the sensitivity of the conclusions to the methods used and choices made.

**3    Detailed comments**

P2 L5: suggest replacing "will likely continue to be so", e.g. for "will likely remain so".

P2 L14-15: I find the distinction between "surface" and "volume" changes a bit confusing as surface losses also consist in changes in volume, just like at the calving front. I suggest replacing "volume losses" for "frontal losses at the terminus of outlet glaciers".

P2 L27: The GBI acronym should not be introduced if not used afterwards.

P3 L1-L2: this sentence while clear is very long, I suggest breaking it in two shorter sentences, e.g. by starting a second sentence instead of "which, in turn" (P2 L30).

P3 L11: This value corresponds to a freshly exposed bare ice at the beginning of the melt season and can decrease by several tens in the next months. As a range of values is provided for snow, I suggest doing so for bare ice too. In Wehrlé et al. (2021) we determined a bare-ice albedo at ice-ablation onset (that we called bare-ice-onset albedo) of 0.565, and a mean minus one standard deviation as low as 0.314, 36 days after bare-ice onset. A range of 0.57-0.31 could therefore be used. Simply specifying the value presented here is at bare-ice onset would also make it more clear.

P3 L13: "encompasses only a small fraction of the GrIS": since a value for the runoff is give, it would be interesting to have an average value for the bare ice area ratio, too.

P3 L18: It could be interesting to give values for the variability of bare ice albedo here, or at least at some point in the introduction. The values reported in Wehrlé et al. (2021) could be used.

P4 L7: I think Stibal et al. (2017) is also important to include, they show ice algae enhance bare ice darkening.

P5 L27: The 70°N restriction should be made clearer. This is not completely clear to me, but as I understand it data above 70°N is included for bare ice extent and disregarded for bare ice albedo and runoff comparisons. This should be clearly stated here.

P7 L13-14: For clarity and to help the reader I suggest adding that, because this is a conservative estimate, it consists in a first simplified estimate of the bare extent which is further refined by the two conditions on snow depth and average density. See next comment.

P7 L13-14: This is an efficient masking for melt years above or close to average, however for cold years part of the bare ice area is probably disregarded right after this first step as the ELA might be higher than the long-term average. I suppose this is not influencing the results a lot, but this should be stated and discussed. This is linked to the first general comment. The influence of the long-term average ELA on the mapping of individual years and its potential limitations are important to further discuss.

P7 L19: suggest explaining very shortly why four different scores are used, why this is needed, and how different they are from each other. I suppose most of the cryosphere community is not necessarily familiar with forecast verification.

P8 L1: I think this statement should be made stronger. Every year, large areas with albedo values below 0.5 are observed. Based on the analysis in Wehrlé et al. (2021), the average albedo from the 20 stations included in the study is below 0.5 for more than a month during the melt season. The surface albedo dropping below 0.5 over the summer can therefore be considered as a common event across the bare ice area.

P8 L21: I suggest adding mean bias and RMSE/RMSD here. A qualitative assessment is given in the next sentence but adding the associated numbers would make the point clearer.

Figure 1: I suggest having the point cloud density as a colormap (instead of black) to get a better sense about the distribution especially at low values where the high densities are saturated.

Figure 2: Non-zero average bare ice days so high in altitude at high latitude (e.g. in the North West) is surprising to me. E.g. on Figure 3 of Wehrlé et al. (2021) (Sentinel-3 data), even for the high melt year 2019, albedo is still high above 0.5/0.6 in these areas. The authors describe this pattern very shortly, but I think the MODIS retrievals alone in those areas deserve a couple more sentences, where I suggest comparing qualitatively to other studies.

P12 L12-13: I suggest reformulating this sentence by starting with the second part, e.g. : "MAR respectively under/overestimating snow melt in colder/warmer years indicates it could be too sensitive to temperature"

P12 L15-16: The authors explain the deviations in average BIE between MODIS retrievals and MAR "stem from the inclusion of 2021 data" within a 20 year data sets which I was a bit surprised about. Indeed, on our side, we haven't detected any major issues with bare ice area retrievals in 2021 using the threshold from Wehrlé et al. (2021) and Sentinel-3 data. The deviations in 2021 BIE retrievals must be very high for the inclusion of the equivalent of ∼5% of the data set size to explain relatively important differences in average BIE. I think a quantification of the BIE differences in 2021 should be included and shortly discussed.

P12 L25: The choice of the threshold on MODIS band 2 might also partly explain this pattern.

P13 L1: Suggest replacing ""eager"" e.g. for "overestimates firn transformation into bare ice"

Figure 4: I suggest adding a panel for the difference between MODIS and MAR results as in Figure 2. MAR bare ice albedo is almost constant, but this would directly make the range of deviation available to the reader.

Figure 5: The sequential blue colormap used in Figure 3 is readable because the order of the curves follows the sequence. However in Figure 5, because the curves are crossing each other, it is getting hard to link them easily to their respective resolutions. Since there is only 4 curves, I suggest using distinct colors.

P16 L15: two times "only", one should be deleted.

P16 L25: suggest adding an average ratio/difference compared to second largest contributor.

P16 L29: suggest adding a range of values.

Figure 6: suggest plotting the data as lines and dots ('o-' e.g. in python). The bars make the visual quite heavy to look at and especially in the case of b), makes the small values close to zero hard to distinguish.

Figure 7: Including September would be interesting to see where is the true maximum of this seasonal trend. Currently, the observed maximum is obtained at the very end of the study period. Interesting patterns -and potentially the highest ratios- might therefore be missed.

P19 L20: suggest including "in central West Greenland" to make it completely clear.

P19 L21-22: I suppose the authors mean South West Greenland by "in the southwest", and not in the southwest of the region of interest (which would be near Greenland's southern tip), as the lowest values on the West coast are determined at the latitude of Disko Island. This was initially misleading to me, I suggest specifying the scale.

P20 L15 P21 L1: This is the kind of thoughts and limitation discussion that I think is needed to include for the spaceborne observations, more specifically for the use of the threshold from Shimada et al, 2016. See general comments.

P21 L10: This is where the uncertainty, or a least the sensitivity of the results on bare ice mapping method could be discussed. It should even deserve a whole paragraph in the discussion in my opinion.

P21 L11-18: Part of the results are not limited below this 70°N limit. As pointed out in an earlier comment, I think this should be made clearer. Whenever the 70°N restriction is mentioned, it is currently presented as if it was applied to any results reported here, but this is not the case.

**References**

Stibal, M., Box, J. E., Cameron, K. A., Langen, P. L., Yallop, M. L., Mottram, R. H., Khan, A. L., Molotch, N. P., Chrismas, N. A. M., Calì Quaglia, F., Remias, D., Smeets, C. J. P. P., Broeke, M. R., Ryan, J. C., Hubbard, A., Tranter, M., As, D., and Ahlstrøm, A. P.: Algae Drive Enhanced Darkening of Bare Ice on the Greenland Ice Sheet, Geophysical Research Letters, 44, 11,463–11,471, https://doi.org/10.1002/2017gl075958, 2017.

Wehrlé, A., Box, J. E., Niwano, M., Anesio, A. M., and Fausto, R. S.: Greenland bare-ice albedo from PROMICE automatic weather station measurements and Sentinel-3 satellite observations, GEUS Bulletin, 47, https://doi.org/10.34194/geusb.v47.5284, 2021.

---

## Author Comment (AC2)

Dear Adrien Wehrlé,

We thank you for your thorough and constructive comments on our paper. All authors have discussed your comments and we have addressed all of them below. Where it is appropriate, we adjusted the text and figures in our manuscript accordingly. We believe these changes have improved the quality and clarity of our manuscript.

Sincerely,
Raf Antwerpen

1. Summary

The authors assess the performances of the Modèle Atmosphérique Régional (MAR) in simulating Greenland bare ice albedo by comparing MAR results over the period 2000-2021 with albedo and bare ice extent retrievals from MODIS optical imagery. Spatial and temporal similarities and deviations in bare ice extent and bare ice albedo between MAR and MODIS retrievals are first presented and discussed. An energy balance model is then used to parameterize meltwater production and present the consequences of significant deviations between modeled and observed bare ice albedo in runoff estimation. The authors have presented a well-written and coherent manuscript on a topic and scope that suits well for The Cryosphere. Reporting on and understanding the performances of regional climate models in simulating bare ice processes and their consequences is indeed a key step to move towards an improved representation of this complex but highly important area of the Greenland ice sheet. By quantifying the deviations between MAR simulations and spaceborne observations, the authors make an essential contribution to the basis of an improved projection of the Greenland ice sheet's future evolution. Along with several minor comments, I nevertheless here point out a few substantial concerns that are in my opinion essential to address before publication. Several choices made by the authors are I think not motivated and discussed enough to give the reader a clear picture about their limitations and potential implications. While I mostly support those choices, it is currently hard to get a sense about the magnitude of their influence on the conclusions. My main concerns are about the choice of method for bare ice mapping with MODIS data, the use of a spatially- and temporally-averaged ELA and a 1999-2002 static ice mask and DEM. These different points are further discussed in the "General comments" section below.

*We appreciate your valued comments on our manuscript.*

2. General comments

The threshold of 0.6 applied to MODIS band 2 has been determined by Shimada et al, 2016 using one single satellite image analyzed over West Greenland. This threshold is here applied by the authors to data covering the entire ice sheet from 2000 to 2021 (with a focus below 70 N for runoff estimations). It is therefore important to discuss a potential lack of representativity of this threshold at the ice sheet scale, and present potential associated limitations. In this way, important to better understand the sensitivity of the results to this threshold: does a small variation in this value have a significant impact on the results presented here? And if so, to which extent?

*We have added the following text to section 4. Discussion (page 23):*
*We recognize that the choice of the threshold in band 2 of MOD09GA to determine the bare ice extent adds additional uncertainty to our results. (Shimada et al., 2016) determined the 0.6 threshold from only one image of southwest Greenland from MODIS' sub-sampled calibrated radiance product MOD02SSH from July 12, 2012 on 5km spatial resolution. The authors picked this image because of the maximum variability in surface conditions within the image. The 0.6 threshold is simply defined as the mean of the spectral reflectance of snow and bare ice in band 2 (841-876 nm). Despite the small range of spatial and spectral data used in defining this threshold, a comparison with a coincident image from Landsat 8/OLI shows a good agreement in surface*

*condition classification. A relative error in bare ice classification of only 0.16% leads us to believe that the threshold of 0.6 for bare ice classification found by (Shimada et al., 2016) is robust. As a sensitivity test, we reprocessed the MOD09GA data using a slightly lower threshold for bare ice classification of 0.55 for the year 2009, whose bare ice extent is representative as an average year in the period 2000-2021. We find that the maximum annual bare ice extent is 17.22% lower if we use a threshold of 0.55 as opposed to 0.6. On average, from June 1 through August 31, the daily bare ice extent is 23.73% lower if we use a threshold of 0.55. This shows that the bare ice extent is sensitive to the choice of the threshold in band 2 of MOD09GA. However, no other estimates for this threshold are available in the current literature. We therefore believe that the 0.6 threshold is currently the best estimate.*

*We did not add this to the manuscript, but it is worth mentioning:*
*We calculated the reflectance distributions of MOD09GA band 2 over the GrIS area below the long-term ELA for 10-day periods in June-August. Figure 1 shows the reflectance distribution for July 10-20, 2012. The bimodal distribution shows a clear distinction between the reflectance of ice and the reflectance of snow with a local minimum of 0.591. The average reflectance value at the local minimum of the bimodal distribution of all the 10-day periods is 0.597 ± 0.0134. This result shows that the 0.6 threshold is a reasonable choice for differentiating bare ice from snow. We excluded June 1-19 from this calculation since no clear bimodal distribution is visible. The bare ice extent during this period is relatively small, making it less important to include in these calculations.*

[Figure]

*Figure 1: Reflectance distribution of MOD09GA band 2 over the GrIS below the long-term ELA for July 10-20, 2012. The orange line denotes the best fit to the histogram.*

*Moreover, we plan on going to Greenland soon to collect in situ spectrometer data, as well as multispectral drone data to create better ground truth data. This allows us to assess the impact of the 0.6 threshold and, if necessary, to find a new, optimal threshold.*

In this direction, and as an example and illustration, I used MOD10A1 (1km resolution) data I had locally on hand for the period 2000-2019 and applied the bare-ice-onset albedo value of 0.565 determined in Wehrlé et al. (2021) to quantify the daily bare ice extent. In Wehrlé et al. (2021),

the seasonal evolution of bare ice albedo across Greenland has been studied using data from 20 automatic weather stations (105 station years). A bare-ice albedo at ice-ablation onset of 0.565 - used here- has been determined (and called bare-ice-onset albedo). The MOD10A1 data used here is gapless, a given pixel value is fixed to the last cloud-free retrieval until the next cloud-free update is available. The static GIMP mask has been used. The data and method used here are different to those presented in this manuscript but both exercises quantify the same variable and therefore, should ideally show low deviations. As visible in the figure below, large variations are shown compared to MOD09GA with the Shimada et al 2016 threshold, and the bare ice area computed here is in much better agreement with MAR. Making the assumption MAR is underestimating runoff over the whole ice sheet (although only shown below 70 N in this study), one could suggest this pattern is mainly due to MAR's incapacity to model low-enough bare ice albedo values (Figure 5b of this study) and not because of low performances in bare ice classification. However, a spatial analysis would be needed to make sure that, more than having similar magnitude, both bare ice extents actually correspond to similar areas.

*We appreciate your commitment to this comment by processing data yourself. We are excited that using the 0.565 threshold on the MOD10A1 product provides an additional method for estimating bare ice extent from satellite imagery. However, we want to emphasize that the MOD10A1 product is less reliable for latitudes higher than 70N (Alexander et al., 2014). We therefore prefer to use the MOD09GA product to determine the bare ice extent from MODIS. We acknowledge that the MOD10A1 product provides a better match with MAR than does the MOD09GA product. We will therefore elucidate our MOD09GA results by plotting the seasonal means of the different groups as noted in Table 1, as well as an additional group that represents pixels that MAR models as bare ice and MODIS observes as clouds. This will better elucidate, on a daily basis, when and how MAR and MOD09GA differ in their bare ice estimates and what role clouds play in this result.*

*We believe the gap filling method is significantly different from the method we use and it is therefore expected that the gap-filled results differ from our results. Since there is currently no way of knowing whether a cloud-obstructed pixel contains snow or ice on the GrIS surface, we prefer to provide a conservative estimate (only the bare ice we actually observe) rather than use the fixed-pixel value of the last cloud-free retrieval, especially since the presence of clouds may be accompanied by precipitation and thus a change in the surface conditions. As a sensitivity test, we processed the MODIS data again using the gap-filling method (Figure 2). Even though the discrepancy between MODIS and MAR decreases for the seasonal bare ice extent (Figure 2b), the discrepancy increases significantly for the maximum annual bare ice extent (Figure 2a). We therefore believe that using this method does not unambiguously decrease the discrepancy between MAR and MODIS. and we prefer to stick with our non-gap-filling method.*

*We discovered a small bug in our code and adjusted it to slightly improve the cloud representation from the MODIS data. With the new cloud uncertainty range, MAR is almost fully incorporated within the MODIS lower and upper estimates for the seasonal bare ice extent (Figure 2b). The cloud uncertainty for the maximum annual bare ice extent (Figure 2a) changed negligibly.*

*In our manuscript we focus on the impact of biases in MAR bare ice albedo on melting, rather than the impact of bare ice extent. We only analyze the pixels that are bare ice in MODIS and*

*MAR simultaneously. Hence, we exclude the possibility of a discrepancy in bare ice extent as a cause for meltwater production underestimation.*

[Figure]

*Figure 2: a) annual maximum bare ice extent in 2000-2021, averaged over JJA and b) seasonal bare ice extent in JJA, averaged over 2000-2021. The red line shows the MODIS bare ice extent using the gap-filling method.*

While MAR uncertainty and potential sources of errors are widely discussed which adds a lot of value to the conclusions drawn in this manuscript, there is currently no similar considerations on the observation side of the study. The example presented here supports the real need for a substantial discussion of the observation retrievals and their sensitivity to the methods applied.

*We have added a paragraph on this to section 4. Discussion (page 23).*

As a first step in the mapping of bare ice, the bare ice extent is bounded to the area below the spatially and temporally-averaged ELA to exclude "sporadically high ablation cell detections". How important are these cells? Do they correspond to <1% of the total bare ice extent or more? Addressing those questions would be necessary to get a better sense about the sensitivity of the results to the use of the long-term average ELA (more details in detailed comment P7 L13-14).

*The SMB is negative for the area below the ELA. However, this does not mean that this area is immediately bare ice. It means, that bare ice can be exposed here. If we use an annually averaged ELA, or even a daily ELA, as simulated by MAR, to determine the annual bare ice extent we rely too much on MAR determining the ELA, making a comparison between MAR and MODIS biased. A long-term average (2000-2021) makes the comparison less dependent on MAR. The standard deviation in annual average ELA is 149.7 m as simulated by MAR between 2000 and 2021.*

*For MAR, the total bare ice extent is 1.1% larger when not using an ELA cutoff. For MODIS, this is 23.7%. The effect of using an ELA is significant for MODIS, though not for MAR. However, we believe that using the ELA cutoff is an important step in providing a proper comparison between MODIS and MAR. Firn and superimposed ice are likely present at elevations near the ELA. We want to focus our comparison on bare ice albedo and meltwater production from the bare ice zone. We therefore believe that performing the analysis on a subsection of the bare ice extent (from using an ELA cutoff) is preferred over performing the analysis on the full bare ice extent plus additional areas covered by firn or superimposed ice (from not using an ELA cutoff), since the latter will make our analysis less purely focused on bare ice.*

*I added to section 2.3 Bare ice extent (page 6):*
*We note that this method may provide a conservative estimate of the bare ice extent during warm, high-melt years, as the ELA in such years may lie at a higher elevation than the long-term average ELA.*

The GIMP ice mask and DEM used in different sections of the study to assess results over the 2000-2021 time period have been created from a combination of Landsat-7 and RADARSAT-1 imagery acquired between 1999 and 2002. The ice sheet evolved significantly in 20 years. It is important to discuss if the use of a static ice mask and DEM has an important impact on the different results it is related to, in order to, here again, be able to get the full picture about the sensitivity of the conclusions to the methods used and choices made.

*We note that the static GIMP ice mask and DEM are constructed from Landsat-7 and RADARSAT-1 imagery acquired between 1999 and 2002, which only overlaps for 3 out of the 22 years of the study period in this study. However, even during the Holocene Climatic Optimum (8-5 ka), when mean annual surface temperatures on Greenland were 2-3° higher than today (Nielsen et al., 2018), the maximum simulated ice margin retreat velocity in southwest Greenland was ~30 m/yr for an extreme climatology (Young et al., 2021). Similar values for Holocene retreat rates in southwest Greenland are shown in (Lesnek et al., 2020). Using this as an analogy for modern ice margin retreat, this could lead to ~660 m of retreat between 2000 and 2021 in an extreme situation, while the ablation zone reaches more than 100 km inland in this area. Hence, this will only lead to a small deviation in the total estimated bare ice extent. We therefore believe that the static GIMP ice mask suffices for the purposes of this study. At elevations near the long-term ELA (1679 m), observed thinning rates estimated using CryoSat-2 data are in the order of ~1 m/yr between 2011 and 2014 (Helm et al., 2014). The use of a static DEM will therefore merely lead to a slightly more conservative estimate of the ablation zone as the ice sheet is actually thinning while we prescribe a static ice sheet elevation.*

*I added this to section 2.3 Bare ice extent (page 7):*

*We note that the static GIMP ice mask and DEM are constructed from Landsat-7 and RADARSAT-1 imagery acquired between 1999 and 2002, which only overlaps for 3 out of the 22 years of the study period in this study. However, the impact on the estimated bare ice extent should be small given estimates for ice margin retreat rates and thinning rates (Helm et al., 2014; Lesnek et al., 2020; Young et al., 2021). We therefore believe that the static GIMP ice mask suffices for the purposes of this study.*

**3. Detailed comments**

P2 L5: suggest replacing "will likely continue to be so", e.g. for "will likely remain so".

*We have adjusted the text as suggested.*

P2 L14-15: I find the distinction between "surface" and "volume" changes a bit confusing as surface losses also consist in changes in volume, just like at the calving front. I suggest replacing "volume losses" for "frontal losses at the terminus of outlet glaciers".

*We have adjusted the text as suggested.*

P2 L27: The GBI acronym should not be introduced if not used afterwards.

*We have adjusted the text as suggested.*

P3 L1-L2: this sentence while clear is very long, I suggest breaking it in two shorter sentences, e.g. by starting a second sentence instead of \which, in turn" (P2 L30).

*We have adjusted the text as suggested.*

P3 L11: This value corresponds to a freshly exposed bare ice at the beginning of the melt season and can decrease by several tens in the next months. As a range of values is provided for snow, I suggest doing so for bare ice too. In Wehrlé et al. (2021) we determined a bare-ice albedo at ice-ablation onset (that we called bare-ice-onset albedo) of 0.565, and a mean minus one standard deviation as low as 0.314, 36 days after bare-ice onset. A range of 0.57-0.31 could therefore be used. Simply specifying the value presented here is at bare-ice onset would also make it more clear.

*We have adjusted the text as suggested.*

P3 L13: "encompasses only a small fraction of the GrIS": since a value for the runoff is give, it would be interesting to have an average value for the bare ice area ratio, too.

*We have adjusted the text as suggested.*

P3 L18: It could be interesting to give values for the variability of bare ice albedo here, or at least at some point in the introduction. The values reported in Wehrlé et al. (2021) could be used.

*We have adjusted the text as suggested.*

P4 L7: I think Stibal et al. (2017) is also important to include, they show ice algae enhance bare ice darkening.

*We have adjusted the text as suggested.*

P5 L27: The 70 N restriction should be made clearer. This is not completely clear to me, but as I understand it data above 70 N is included for bare ice extent and disregarded for bare ice albedo and runoff comparisons. This should be clearly stated here.

*We have adjusted the text as suggested.*

P7 L13-14: For clarity and to help the reader I suggest adding that, because this is a conservative estimate, it consists in a first simplified estimate of the bare extent which is further refined by the two conditions on snow depth and average density. See next comment.

*We have adjusted the text as suggested.*

P7 L13-14: This is an efficient masking for melt years above or close to average, however for cold years part of the bare ice area is probably disregarded right after this first step as the ELA might be higher than the long-term average. I suppose this is not influencing the results a lot, but this should be stated and discussed. This is linked to the first general comment. The influence of the long-term average ELA on the mapping of individual years and its potential limitations are important to further discuss.

*See general comments.*

P7 L19: suggest explaining very shortly why four different scores are used, why this is needed, and how different they are from each other. I suppose most of the cryosphere community is not necessarily familiar with forecast verification.

*We have adjusted the text as suggested.*

P8 L1: I think this statement should be made stronger. Every year, large areas with albedo values below 0.5 are observed. Based on the analysis in Wehrlé et al. (2021), the average albedo from the 20 stations included in the study is below 0.5 for more than a month during the melt season. The surface albedo dropping below 0.5 over the summer can therefore be considered as a common event across the bare ice area.

*We have adjusted the text as suggested.*

P8 L21: I suggest adding mean bias and RMSE/RMSD here. A qualitative assessment is given in the next sentence but adding the associated numbers would make the point clearer.

*We have adjusted the text as suggested.*

Figure 1: I suggest having the point cloud density as a colormap (instead of black) to get a better sense about the distribution especially at low values where the high densities are saturated.

*We have adjusted the figure as suggested.*

Figure 2: Non-zero average bare ice days so high in altitude at high latitude (e.g. in the North West) is surprising to me. E.g. on Figure 3 of Wehrlé et al. (2021) (Sentinel-3 data), even for the high melt year 2019, albedo is still high above 0.5/0.6 in these areas. The authors describe this pattern very shortly, but I think the MODIS retrievals alone in those areas deserve a couple more sentences, where I suggest comparing qualitatively to other studies.

*We have added this to section 3.1 Bare ice extent (page 11):*
*The number of bare ice days observed by MODIS show slightly more inland variation, especially in the northern regions of the ice sheet. For instance, a large circular feature in the northeast reveals that bare ice is exposed for up to 12 days on average during JJA. The geothermal heat flux map of Greenland created by (Martos et al., 2018) shows a circular feature of similar size in the same area with an enhanced heat flux. Increased heat flux from the bedrock to the ice sheet surface could lead to higher ice sheet surface temperatures, which enhances bare ice exposure. This pattern is not captured by the MAR simulations.*

P12 L12-13: I suggest reformulating this sentence by starting with the second part, e.g.: "MAR respectively under/overestimating snow melt in colder/warmer years indicates it could be too sensitive to temperature".

*We have adjusted the text as suggested.*

P12 L15-16: The authors explain the deviations in average BIE between MODIS retrievals and MAR "stem from the inclusion of 2021 data" within a 20 year data sets which I was a bit surprised about. Indeed, on our side, we haven't detected any major issues with bare ice area retrievals in 2021 using the threshold from Wehrlé et al. (2021) and Sentinel-3 data. The deviations in 2021 BIE retrievals must be very high for the inclusion of the equivalent of 5% of the data set size to explain relatively important differences in average BIE. I think a quantification of the BIE differences in 2021 should be included and shortly discussed.

*We have added this to section 3.1 Bare ice extent (page 14):*
*Excluding the 2021 MODIS data provides a trend of 2.272 km²/yr, which is closer to the trend simulated by MAR.*

P12 L25: The choice of the threshold on MODIS band 2 might also partly explain this pattern.

*We have adjusted the text as suggested.*

P13 L1: Suggest replacing ""eager"" e.g. for "overestimates firn transformation into bare ice"

*We have adjusted the text as suggested.*

Figure 4: I suggest adding a panel for the difference between MODIS and MAR results as in Figure 2. MAR bare ice albedo is almost constant, but this would directly make the range of deviation available to the reader.

*We have adjusted the figure as suggested.*

Figure 5: The sequential blue colormap used in Figure 3 is readable because the order of the curves follows the sequence. However in Figure 5, because the curves are crossing each other, it is getting hard to link them easily to their respective resolutions. Since there is only 4 curves, I suggest using distinct colors.

*To keep the aesthetics of the paper consistent, we prefer to keep the colors as is in Figure 5. Since the differences in albedo values between the MAR resolutions are so small, we believe it is not important to be able to distinguish the separate lines.*

P16 L15: two times "only", one should be deleted.

*We have adjusted the text as suggested.*

P16 L25: suggest adding an average ratio/difference compared to second largest contributor.

*We have adjusted the text as suggested.*

P16 L29: suggest adding a range of values.

*We have adjusted the text as suggested.*

Figure 6: suggest plotting the data as lines and dots ('o-' e.g. in python). The bars make the visual quite heavy to look at and especially in the case of b), makes the small values close to zero hard to distinguish.

*We have noted this comment, but we prefer to keep Figure 6 in this style.*

Figure 7: Including September would be interesting to see where is the true maximum of this seasonal trend. Currently, the observed maximum is obtained at the very end of the study period. Interesting patterns -and potentially the highest ratios- might therefore be missed.

*We agree that including September in the analysis could potentially elucidate extended patterns of bare ice extent, albedo and meltwater production biases. However, the uncertainty of MODIS data increases because the solar zenith angle at this latitude during this time of the year increases significantly. MODIS data with a solar zenith angle over 55 degrees show a drop in uncertainty (van Dalum et al., 2020), and the average solar zenith angle in September is higher than 55 degrees for the entire Greenland ice sheet. We therefore believe that adding September to the*

*analysis will significantly increase the uncertainty of the results and we prefer to keep our analysis to JJA. Moreover, using JJA is a standard approach to researching seasonal trends.*

P19 L20: suggest including "in central West Greenland" to make it completely clear.

*We have adjusted the text as suggested.*

P19 L21-22: I suppose the authors mean South West Greenland by "in the southwest", and not in the southwest of the region of interest (which would be near Greenland's southern tip), as the lowest values on the West coast are determined at the latitude of Disko Island. This was initially misleading to me, I suggest specifying the scale.

*We have adjusted the text as suggested.*

P20 L15 P21 L1: This is the kind of thoughts and limitation discussion that I think is needed to include for the spaceborne observations, more specifically for the use of the threshold from Shimada et al, 2016. See general comments.

*We have noted this comment and addressed it in the general comments.*

P21 L10: This is where the uncertainty, or at least the sensitivity of the results on bare ice mapping method could be discussed. It should even deserve a whole paragraph in the discussion in my opinion.

*We have adjusted the text as suggested.*

P21 L11-18: Part of the results are not limited below this 70 N limit. As pointed out in an earlier comment, I think this should be made clearer. Whenever the 70 N restriction is mentioned, it is currently presented as if it was applied to any results reported here, but this is not the case.

*We have adjusted the text as suggested.*